# Chagas Disease-Related Mortality in Spain, 1997 to 2018

**DOI:** 10.3390/microorganisms9091991

**Published:** 2021-09-20

**Authors:** Jose-Manuel Ramos-Rincon, Jara Llenas-García, Hector Pinargote-Celorio, Veronica Sánchez-García, Philip Wikman-Jorgensen, Miriam Navarro, Concepción Gil-Anguita, Violeta Ramos-Sesma, Diego Torrus-Tendero

**Affiliations:** 1Internal Medicine Department, Alicante Institute of Sanitary and Biomedical Research (ISABIAL), Alicante General University Hospital, 03010 Alicante, Spain; hectorpinargote@gmail.com; 2Clinical Medicine Department, Miguel Hernández University of Elche, 03550 Alicante, Spain; 3Internal Medicine Department, Hospital Vega Baja, Foundation for the Promotion of Health and Biomedical Research of the Valencia Region (FISABIO), 03314 Alicante, Spain; jarallenas@gmail.com; 4Dermatology Service, Alicante Institute of Sanitary and Biomedical Research (ISABIAL), Alicante General University Hospital, 03010 Alicante, Spain; verosg1995@gmail.com; 5Internal Medicine Department, Foundation for the Promotion of Health and Biomedical Research of the Valencia Region (FISABIO), University Hospital of Sant Joan, 03550 Alicante, Spain; wikman_phi@gva.es; 6Epidemiology Unit, Public Health Center of Elche, 03302 Alicante, Spain; navarro_mirbel@gva.es; 7Department of Public Health, Science History and Gynecology, Miguel Hernández University of Elche, 03550 Alicante, Spain; 8Internal Medicine Department, Hospital Marina Baixa—Foundation for the Promotion of Health and Biomedical Research of the Valencia Region (FISABIO), 03570 Alicante, Spain; cgilanguita@gmail.com; 9Internal Medicine Service, HLA Inmaculada Hospital, 18004 Granada, Spain; vramossesma@gmail.com; 10Infectious Diseases Unit, Alicante Institute of Sanitary and Biomedical Research (ISABIAL), Alicante General University Hospital, 03010 Alicante, Spain; torrus_die@gva.es; 11Parasitology Area, Miguel Hernández University of Elche, 03550 Alicante, Spain

**Keywords:** Chagas disease, mortality, aged, male, female, HIV infection, cardiopathy, neoplasm, Spain

## Abstract

Background. Chagas disease (CD) is associated with excess mortality in infected people in endemic countries, but little information is available in non-endemic countries. The aim of the study was to analyze mortality in patients admitted to the hospital with CD in Spain. Methods. A retrospective, observational study using the Spanish National Hospital Discharge Database. We used the CD diagnostic codes of the 9th and 10th International Classification of Diseases to retrieve CD cases from the national public registry from 1997 to 2018. Results. Of the 5022 hospital admissions in people with CD, there were 56 deaths (case fatality rate (CFR) 1.1%, 95% confidence interval (CI) 0.8%, 1.4%), 20 (35.7%) of which were considered directly related to CD. The median age was higher in those who died (54.5 vs. 38 years; *p* < 0.001). The CFR increased with age, peaking in the 70–79-year (7.9%, odds ratio (OR) 6.27, 95% CI 1.27, 30.90) and 80–89-year (16.7%, OR 14.7, 95% CI 2.70, 79.90) age groups. Men comprised a higher proportion of those who died compared to survivors (50% vs. 22.6%; *p* < 0.001). Non-survivors were more likely to have neoplasms (19.6% vs. 3.4%; *p* < 0.001), heart failure (17.9% vs. 7.2%; *p* = 0.002), diabetes (12.5% vs. 3.7%; *p* = 0.001), chronic kidney failure (8.9% vs. 1.6%; *p* < 0.001), and HIV (8.9% vs. 0.8%; *p* < 0.001). In the multivariable analysis, the variables associated with mortality were age (adjusted OR (aOR) 1.05; 95% CI: 1.03, 1.07), male sex (aOR 1.79, 95% CI 1.03, 3.14), cancer (aOR: 4.84, 95% CI 2.13, 11.22), and HIV infection (aOR 14.10 95% CI 4.88, 40.73). Conclusions. The case fatality rate of CD hospitalization was about 1%. The mortality risk increased with age, male sex, cancer, and HIV infection.

## 1. Introduction

Chagas disease (CD), or American trypanosomiasis, is a chronic systemic parasitic infection caused by the protozoan *Trypanosoma cruzi*. It is endemic to 21 countries in continental Latin America, with an estimated 6 to 7 million people infected worldwide [1,2]. Spain is a country where immigration, especially from Latin America, has changed the demographic landscape in the last two decades. According to the Spanish National Institute of Statistics, the number of foreigners from Central and South America is around 3.1 million people, or 6.5% of the Spanish population. This migratory phenomenon is not unique to Spain; it also occurs across North America, Europe, Asia, and Oceania and has led to CD becoming a global health problem [3]. At present, an estimated 10% of the world’s CD cases occur outside Latin America. After the USA, Spain is the non-endemic country with the highest number of CD cases and accounts for 75% of all cases in the European Union, with an estimated 65,000 people affected [3].

The clinical course of CD can be divided into an acute phase, an indeterminate phase (where the patient is asymptomatic before the onset of chronic disease symptoms), and a chronic symptomatic phase [1,2]. The interval between infection and onset of the symptomatic phase ranges from 10 to 30 years [4]. Around 20% to 30% of infected people develop heart disease, which is associated with increased mortality [5,6]. In a study of blood donors in Brazil, authors found a rate of disease progression of 1.85% per year [7,8]. CD is considered a neglected tropical disease, with a large loss of total disability-adjusted life years (DALYs) [9]. Moreover, Capuani et al. reported an excess mortality in CD-seropositive compared to CD-seronegative blood donors in Brazil [10], while a recent meta-analysis showed significant excess mortality due to CD, shared among both symptomatic and asymptomatic people with CD compared with non-chagasic individuals [11].

Little information is available on mortality in people with CD living in non-endemic countries. A recent review by Velasco et al. [12] identified only two series of patients in Spain reporting two deaths from CD, both due to sudden death [13,14]. Causes of death in patients with CD in non-endemic countries have not been well studied.

The Spanish National Hospital Discharge Database (SNHDD) belongs to the Spanish Ministry of Health and includes data from all patients discharged from public and private hospitals [15,16]. Several studies have assessed the validity of the SNHDD for conditions including parasitological diseases [17,18,19]. To the best of our knowledge, there is only one previous study analyzing factors associated with CD hospitalization in Spain using the SNHDD, from 2000 to 2011 [20]. We extracted data from 1997 to 2018 with the aim of analyzing mortality, causes of death, and associated factors in patients admitted to hospital with CD.

## 2. Materials and Methods

### 2.1. Data Base and Extractionn of Variables

We conducted an observational retrospective study using the SNHDD [15,16]. This database includes all patients discharged from hospitals/clinics across the country since the early 1990s [18]. Data collected include the sex, age, dates of admission and discharge, days of hospitalization, admission service, circumstances of the hospital discharge, up to 14 discharge diagnoses (main diagnosis, and secondary diagnoses), and up to 20 procedures performed during the hospitalization. Our study period spanned from 1 January 1997 to 31 December 2018 (22 years). Patients’ country of origin was available only from January 2015 [16,21]. The “primary/main diagnosis” refers to the condition that, at discharge, is considered the cause of hospital admission. “Secondary diagnoses” are the diagnoses that coexist with the main diagnosis at the time of admission or during the hospital stay. The circumstances of hospital discharge refer to the final hospitalization outcome: voluntary discharge, transfer to another center, or death.

The criteria for diseases and procedures were defined according to the classifications used by the SNHDD: from 1997 to 2015, the International Classification of Diseases-Ninth Revision, Clinical Modification (ICD-9-CM), and the ICD-10-CM thereafter. We selected hospital admissions according to the codes related to Chagas disease and retrieved data about specific comorbidities, including diabetes mellitus, heart failure, ischemic cardiopathy, cerebrovascular diseases, chronic lung diseases, chronic kidney failure, neoplasms, lymphoma/leukemia, and human immunodeficiency virus (HIV) (Appendix A).

All-cause in-hospital mortality in CD was expressed as the case fatality rate (CFR), or the proportion of in-hospital deaths in relation to the total number of patients hospitalized with CD. In patients who died, we reviewed all the registered diagnoses and classified the deaths as: (1) attributable to CD (directly related, root cause), including (1.A) “Chagas disease with heart involvement” OR “Chagas disease with involvement of another organ” OR “Chagas disease with digestive system involvement unspecified” in the primary diagnosis (1.B): “Cardiac arrest “OR “Cardiogenic shock” OR “Primary cardiomyopathy” OR “unspecified congestive heart failure” in a primary diagnosis AND “Chagas disease with heart involvement” OR “Chagas disease with involvement of another organ” in the second or third diagnoses; (2) not directly attributable to CD, including the rest of patients. Two independent researchers classified deaths according to these categories, involving a third researcher in case of disagreement.

### 2.2. Statistics Analysis

The incidence rates of admissions with CD were calculated per 10^7^ hospitalizations (obtained from SNHDD [15,16]) and per 10^5^ people from Central and South American countries (excluding Caribbean countries), as obtained from the Spanish National Institute of Statistics (Madrid, Spain) [22]. Patients’ epidemiological and clinical characteristics were analyzed using descriptive statistics. Non-parametric continuous variables (as assessed by one-sample Kolmogorov–Smirnov test) are expressed as medians and interquartile ranges (IQR), while categorical variables are expressed as absolute values and percentage. Bivariable comparisons of quantitative and qualitative variables were performed using the Mann–Whitney U-test and the Chi^2^ test or Fisher’s exact test, respectively. A bivariable sensitivity analysis excluding obstetrics/gynecology admission was performed, because there were no fatalities in obstetrics/gynecology services.

All tests were two-tailed, and only *p*-values of less than 0.05 were considered significant. The measure of association was presented as odds ratios (OR) with their 95% confidence intervals (CIs). Multivariable logistic regression analysis was used to identify independent predictors of mortality. Demographic variables, comorbidities, and diagnostic variables yielding a *p* value of less than 0.1 in the univariable analysis were entered into a multivariable logistic regression, using a stepwise selection method with the likelihood ratio test.

Model validity was evaluated using the Hosmer–Lemeshow test for estimating goodness of fit to the data and its discriminatory ability using the area under the curve (AUC). The regression analysis values were expressed as adjusted ORs and 95% CIs. All statistical analyses were performed using the IBM SPSS package for Windows v25.0 (IBM Corp, Armonk, NY, USA).

### 2.3. Ethical Aspects

This study involves the use of medical data from the SNHDD. To guarantee patients’ anonymity, the database was provided to us by the Spanish Ministry of Health after removal of all potential patient identifiers. According to the confidentiality agreement with the Ministry, researchers cannot provide the data to other researchers, so other researchers must request the data directly from the Ministry. The study protocol was approved by the Clinical Research Ethics Committee of the Alicante General University Hospital (Alicante, Spain) (Ref. CEIm: PI2021-102). The procedures described here were carried out in accordance with the ethical standards described in the Revised Declaration of Helsinki in 2013.

## 3. Results

### 3.1. CD Hospitalization and Incidence Rate

During the study period, the SNHDD included data for a total of 79,915,683 admissions. Of these, 5022 were in patients with CD (22.9% males and 77.1% females). The number of yearly admissions increased over the study period, from 8 cases in 1997 to 558 in 2018. Thus, in the first half of the study period (1997–2008), there were a total of 643 CD admissions, compared to 4379 in 2009–2018 (Figure 1).

The overall incidence rate was 62.8 cases per 10^7^ hospitalizations and ranged annually from 2.7 × 10^7^ hospitalizations in 1997 to 124.1 × 10^7^ hospitalizations in 2018. With regard to the CD hospitalization rate in Latin-American-born residents in Spain, this was 13.7 × 10^5^ admissions overall and ranged annually from 0.7 × 10^5^ admissions in 1998 to 22.2 × 10^5^ admissions in 2018. Figure 2a,b shows the incidence rate per 10^7^ hospitalizations and per 10^5^ Latin-American-born residents in Spain, respectively.

### 3.2. Fatality and Case Fatality Rate

Of the 5022 CD admissions, 56 patients died (CFR 1.1%, 95% CI 0.9, 1.5), 11 in the 1997–2008 period (CFR 1.7%, 95% CI 0.9, 3.1) and 45 in 2009–2018 (CFR 1.0%, 95% CI 0.8–1.4), with no significant differences in the CFR between periods (*p* = 0.52). Table 1 shows the annual rates.

The CFR increased with age and was especially pronounced in the 70–79-year (CFR 7.9%, 95% CI 0.4, 15.4; OR 6.27, 95% CI 1.27, 30.90) and 80–89-year (CFR 16.7%, 95% CI 0.7, 33.5; OR 14.70, 95% CI 2.70, 79.90) age groups, as shown in Table 2. The CFR for CD admissions was less than for non-CD admissions in Spain (CFR 4.1%, 3,170,961 deaths/77,729,072 hospitalizations). Appendix A compares the CFR between CD vs. non-CD admissions by study year.

Of the 56 deaths, 20 (35.7%) were considered directly related to CD. Appendix A lists details for these patients. Appendix A shows the age, sex, diagnosis, and length of admission for the 36 cases of non-CD death.

### 3.3. Risk Factorrelated with Mortality

Demographic and clinical characteristics of surviving vs. non-surviving inpatients with CS are shown in Table 3. Median age was higher in CD patients who died compared to those who survived (54.5 years vs. 38 years; *p* < 0.001). Half the patients who died were men, compared to just 22.6% of survivors (*p* < 0.001). Compared to survivors, a higher proportion of non-survivors had neoplasms (19.6% vs. 3.4%; *p* < 0.001), chronic heart failure (17.9% vs. 7.2%; *p* = 0.002), diabetes (12.5% vs. 3.7%; *p* = 0.001), chronic kidney failure (8.9% vs. 1.6%; *p* < 0.001), HIV (8.9% vs. 0.8%; *p* < 0.001), and lymphoma/leukemia (5.4% vs. 1.0%; *p* = 0.02).

Most of the non-survivors were admitted in medical services 74.8% (*n* = 43), while most survivors (59.1%) were admitted to gynecology or obstetrics services (*p* < 0.001). Half the non-survivors were admitted for CD with heart involvement, compared to 29.9% of survivors (*p* < 0.001). Regarding CD with involvement of other organs, more survivors compared to non-survivors were admitted for CD without organ involvement (57.1% vs. 35.7%, *p* < 0.001)

Data on country of birth were available in 1211 patients, most of whom were from Bolivia (59.0%). Mortality was similar across groups (Table 3). The median length of stay was higher in non-survivors than survivors (10 days vs. 3 days; *p* < 0.001).

Table 4 shows the results of the multivariable analysis of risk factors associated with mortality in hospitalized CD patients. Independent risks for death in patients with CD were age (adjusted OR (aOR) 1.05), male sex (aOR 1.79), cancer (aOR 4.89), and HIV (aOR 14.10). In this model, the *p* value for the Hosmer–Lemeshow goodness-of-fit test was 0.121, with an AUC of 0.821 (95% CI 0.751, 0.891).

In a sensitivity analysis, we excluded the 2993 patients admitted in the obstetrics/gynecology services. That analysis confirmed that non-survivors were older, had longer admissions, were more likely to be admitted in medical wards, and had more comorbidities (chronic kidney failure, neoplasms and HIV) compared to survivors (Table 4). Multivariable analysis showed that age (aOR 1.04), cancer (aOR 4.46) and HIV infection (aOR 10.4) were independently associated with non-survival in these patients.

## 4. Discussion

This is the first series to report in-hospital mortality of patients with CD in a non-endemic country. The results highlight a consistent CFR of approximately 1%. Mortality was higher in older people, men, and people with cancer or HIV. Approximately a third of the deaths were directly attributable to CD.

Regarding the CD-related mortality in Latin America, there is scant epidemiological information from endemic countries [23]. Some information is available from different series of cohorts and mortality registries, where a range of baseline aspects have been evaluated in various countries. These studies have analyzed different aspects such as rates as well as temporal and geospatial trends of mortality due to CD in general or specifically chagasic cardiomyopathy [23,24,25,26,27]. Other studies analyze excess preventable mortality from the disease and the geo-economic implications of the problem [11].

For example, Olivera et al. [21] analyzed epidemiological characteristics and temporal trends in CD-related mortality in Colombia from 1979 to 2018, using death records and population data from the National Administrative Department of Statistics (using ICD-9 and ICD-10 codes). Of the 7,287,461 deaths recorded, 3,276 (0.04%) were related to CD: 2827 (86.3%) as an underlying cause and 449 (13.7%) as an associated cause.. 

The average annual age- and sex-standardized mortality rate showed a significant upward trend. The highest rates of CD-related death were in men. Our results from a non-endemic country are similar, showing growth in absolute mortality and the highest CD death in men.

In Latin America, premature mortality due to Chagas is responsible for increasing productivity losses. An investigation in Colombia from 2010 to 2017 studied 1261 deaths due to CD, reporting that premature deaths from Chagas resulted in 48,621 potential years of work lost and a cost of USD 29 million (current values of lost lifetime income [24]). While assessing productivity losses was beyond the scope of the study, we observed that 32 of the 56 patients who died (57.1%) were under 60 years old; thus, in line with an increased loss in productive life-years. Regarding the analysis of mortality in endemic areas in Latin America, several studies in Brazil, Colombia, Chile and Ecuador have examined the spatial patterns of CD mortality, enabling the identification of priority areas for health services planning and control strategies [25,26,28,29].

Most studies analyzing mortality in CD patients have reported that mortality from CD—as with most diseases—increases with age [23,25,26,27,29]. In our study, non-survivors were older, with the highest mortality in patients aged 70–79 years and 80–89 years. Migratory flows are dominated by younger adults, so the number of older immigrants is relatively modest; however, mortality was much higher in these groups.

Moreover, CD-associated mortality was higher in men than in women, which is consistent with several series in endemic countries [23,25,27,29]. In our study, in hospitalized patients with CD in non-endemic country, half of non-survivors were men, compared to just 22.9% of all the CD inpatients. This imbalance can be attributed to the widespread prenatal screening programs for Chagas disease and the fact that most included women in our study were admitted to give birth [11]. Additionally, more Latin American women compared to men live in Spain [22].

Chagas cardiomyopathy is the main cause of death in CD patients [30], with several Brazilian studies investigating predictive factors related to this pathology. In 2019, Gali et al. [31] included patients with Chagas cardiomyopathy, finding that those who received implantable cardioverter-defibrillators, older people (≥65 years), and people with left ventricular dysfunction (LVEF < 35%) had a worse prognosis. Sherbuk et al. [32] highlighted the impact of severe Chagas cardiomyopathy on short-term mortality in these patients. Moreover, the presence of decreased ejection fraction and other clinical signs of congestive heart failure have added predictive value for mortality. Nadruz et al. [33] showed that the absolute death rates decreased over time in patients with Chagas and non-Chagas cardiomyopathies, but mortality increased in patients with heart failure. In our study, it was not possible to analyze the different aspects of Chagas cardiomyopathy. However, 35.7% of our inpatient population died with cardiac involvement (“Chagas disease with heart involvement” or “cardiac arrest” or “cardiogenic shock” or “unspecified congestive heart failure”).

CD is also associated with ischemic stroke [8,34] and mortality from this cause. Again in Brazil, Montanero et al. [35] found that 10% of people with CD and ischemic stroke died. In our study, 5% of patients had a secondary diagnosis of cerebrovascular disease, while one in five had a diagnosis for CD with involvement of an organ other than the heart.

In several studies of CD mortality in Latin America, most deaths in people with CD are attributed to that cause, but one in five are associated with a different cause of death [23,36]. In contrast, just a third of the deaths in our study were considered to be directly related to CD. Of those associated with other diseases, the main cause of death was neoplasms (including lymphoma/leukemia), which is consistent with the main cause of non-CD death among people with CD in epidemiological studies in endemic countries [35]. Our multivariable analysis showed that neoplasms were associated with mortality in people with CD, while cardiopathy was not. These results are different from other series in endemic countries [32,33,34,35]. It can be a different treatment of cardiopathy available in our country as transplantation or use of Internal atrial defibrillation.

CD is an opportunistic infection in people living with HIV/AIDS. Patients with HIV and chronic *T. cruzi* infection, when severely immunosuppressed, may experience a reactivation, most commonly manifested by meningoencephalitis and sometimes intracerebral chagomas. The second most common manifestation is myocarditis. In our study, about 10% of non-survivors with CD also had an HIV co-infection. This overlap is frequent in endemic as well as in non-endemic areas [37,38,39]. Martins-Melo et al. found that acute CD with cardiac involvement and chronic CD affecting the nervous system were the most common conditions in people with CD and HIV in Brazil. In our study, three in five patients who died had central nervous system involvement (toxoplasmosis or hemiplegia). As we had no access to clinical records, we were not able to assess the lesions coded as toxoplasmosis, which could well, in fact, have been cerebral chagomas stemming from CD reactivations in HIV-positive patients.

Moreover, CD admissions show a significant upward trend in Spain starting from 2006. This is due to a sharp increase in immigration from Latin America to Spain, which started in 1997 but accelerated from 2005 to 2010, especially in women of childbearing age. Migration in women of childbearing age from Latin America may spread CD to non-endemic areas through vertical transmission [40]. If CD is diagnosed during pregnancy, the transmission of *T. cruzi* to the child cannot be prevented, so early diagnosis in newborns is essential for administering appropriate treatment [40,41]. In Spain, serological screening of *T. cruzi* infection in pregnant women from Latin American is not systematically carried out, but depends on each autonomous community; some regions have implemented prenatal screening programs since 2007 [42,43]. Moreover, a community-based intervention implemented in Spain has shown to be effective in providing access to Chagas disease diagnosis, which has increased in recent years [44,45].

One strength of this study is that it is the first to our knowledge that analyzes in-hospital mortality in patients with CD in a non-endemic country. However, the study also has some limitations. First, we used secondary data from a hospital register, and researchers do not have access to the complete medical records of the patients, and the classification code of diseases is sometimes not easy to perform and can lead to some error. Second, the number of deaths related to CD is underestimated, because we do not include deaths occurring outside the hospital (death certificates). Third, this was a retrospective study, so we had no opportunity to review patients’ medical histories, which would have allowed us to check data for accuracy (quality of information on causes of death). Fourth, the country of birth of hospitalized patients was not recorded until recently, so we did not perform an in-depth analysis of this variable. Moreover, the database included the country of birth, but in the data obtained, a third were recorded as being from Spain, which is probably due to clinicians’ interpreting this variable as the patients’ nationality (i.e., in Latin-American-born people naturalized as Spanish).

## 5. Conclusions

On the other hand, we included multiple causes of death to detect the disease, and we consider the results of this study to be valid and representative of CD-associated mortality in Spain (non-endemic country). Overall, our mortality analysis indicates that mortality in people with CD is related to older age, male sex, cancer, and HIV infection, as in other diseases. These results highlight the importance of continuing hospital-based surveillance of CD in non-endemic countries.

## Figures and Tables

**Figure 1 microorganisms-09-01991-f001:**
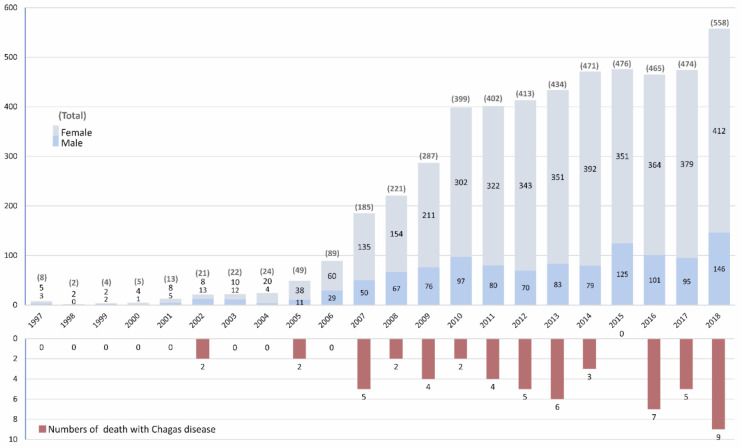
Number of hospitalizations by sex and in-hospital mortality in people with Chagas disease.

**Figure 2 microorganisms-09-01991-f002:**
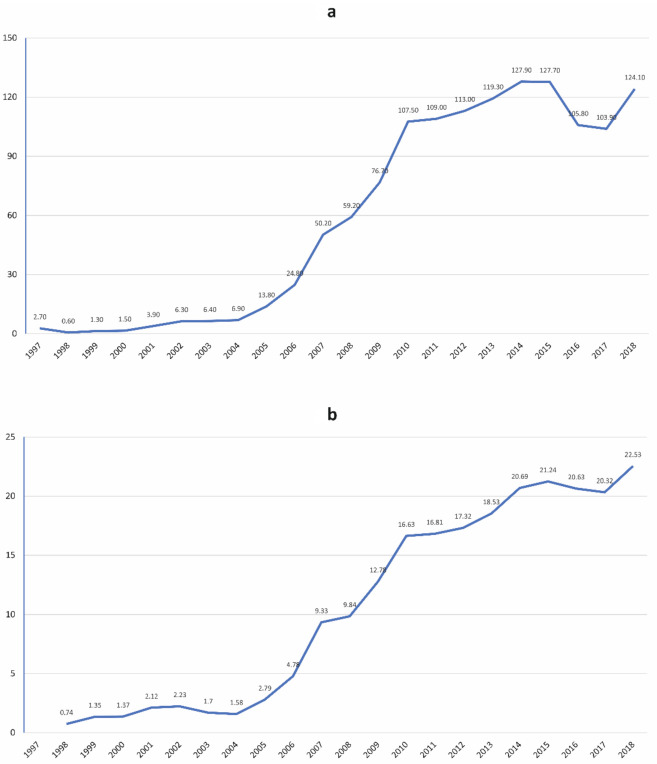
Incidence rates of hospitalizations with CD per (**a**) 10^7^ hospitalizations and (**b**) 10^5^ people from Central and South America.

**Table 1 microorganisms-09-01991-t001:** Case fatality rate (CFR) in hospitalized people with Chagas disease, 1997–2018.

Year	No. Deaths/No. Cases	% of Total Death	CFR % (95% CI)
1997	0/8	0.0	0.0 (0–32.4)
1998	0/2	0.0	0.0 (0–65.7)
1999	0/4	0.0	0.0 (0–48.6)
2000	0/5	0.0	0.0 (0–43.6)
2001	0/13	0.0	0.0 (0–22.8)
2002	2/21	3.6	9.5 (2.6–28.9)
2003	0/22	0.0	0.0 (0–14.7)
2004	0/24	0.0	0.0 (0–13.8)
2005	2/49	3.6	4.1 (1.1–13.7)
2006	0/89	0.0	0.0 (0–4.1)
2007	5/185	8.9	2.7 (1.1–6.2)
2008	2/221	3.6	0.9 (0.3–3.0)
2009	4/287	7.1	1.4 (0.5–3.5)
2010	2/399	3.6	0.5 (0.1–1.8)
2011	4/402	7.1	1.0 (0.4–2.5)
2012	5/413	8.9	1.2 (0.5–2.8)
2013	6/434	10.7	1.4 (0.6–2.9)
2014	3/471	5.4	0.6 (0.2–1.8)
2015	0/476	0.0	0.0 (0–0.8)
2016	7/465	12.5	1.5 (0.7–3.1)
2017	5/474	8.9	1.1 (0.5–2.4)
2018	9/558	16.1	1.6 (0.8–3.1)
Total	56/5022	100	1.1. (0.9–1.5)

CI: confidence interval.

**Table 2 microorganisms-09-01991-t002:** Case fatality rate (CFR) in hospitalized people with Chagas disease, by age group.

Age in Years	No. Deaths/ No. Cases	% of Total Deaths	CFR (95% CI)	OR (95% CI)	*p*-Value
0–9	2/149	3.6	1.3 (0.4–4.7)	1	-
10–19	1/50	1.8	2.0 (0.4–10.5)	1.50 (1.13–16.90)	0.74
20–29	2/699	3.6	0.3 (0.08–1.0)	0.21 (0.03–1.50)	0.12
30–39	7/1888	12.5	0.4 (0.2–0.8)	0.27 (0.56–1.32)	0.11
40–49	8/1086	14.3	0.7 (0.4–1.5)	0.54 (0.11–2.59)	0.45
50–59	12/684	21.4	1.8 (1.0–3.0)	1.31 (0.29–5.92)	0.72
60–69	11/340	19.6	3.2(1.8–5.7)	2.45 (0.53–11.22)	0.25
70–79	7/89	12.5	7.9 (0.4–15.4)	6.27 (1.27–30.90)	**0.024**
80–89	5/30	8.9	16.7 (0.7–33.5)	14.7 (2.70–79.90)	**0.002**
90–100	1/7	1.8	14.3 (2.6–51)	12.50 (0.98–154.12)	0.53

CI: confidence intervals; OR: odds ratio. Statistically significant *p* values in bold.

**Table 3 microorganisms-09-01991-t003:** Demographic and clinical characteristics of surviving vs. non-surviving inpatients with Chagas disease (CD).

Demographic Variables	Total Cases			Non-Obstetrics/Gynecology Cases	
	Non-Survivors (*n* = 56)	Survivors (*n* = 4966)	*p*-Value	Survivors (*n* = 2937)	*p*-Value
Age, median (IQR)	54.5 (41, 68)	38 (31,48)	**<0.001**	45 (36, 54)	**<0.001**
Men, *n* (%)	28 (50)	1121 (22.6)	**<0.001**	1816 (61.8)	0.071
Comorbidities, *n* (%)					
Neoplasms	11 (19.6)	169 (3.4)	**<0.001**	152 (5.2)	**<0.001**
Heart failure	10 (17.9)	359 (7.2)	**0.002**	357 (12.2)	0.20
Diabetes mellitus	7 (12.5)	186 (3.7)	**0.001**	182 (6.2)	0.055
Chronic kidney failure	5 (8.9)	80 (1.6)	**<0.001**	78 (2.7)	**0.018**
HIV infection	5 (8.9)	40 (0.8)	**<0.001**	36 (1.2)	**<0.001**
Chronic lung diseases	4 (7.1)	140 (2.8)	0.054	111 (3.8)	0.20
Ischemic cardiopathy	4 (7.1)	165 (3.3)	0.12	161 (5.5)	0.59
Cerebrovascular diseases	3 (5.4)	130 (2.6)	0.20	127 (4.3)	0.71
Lymphoma/leukemia	3 (5.4)	52 (1.0)	**0.023**	52 (1.8)	0.082
Hospital admission, *n* (%)					
Obstetrics/gynecology Wards	0 (0)	2029 (40.9)	**<0.001**	-	-
Medical wards	48 (85.7)	1758 (35.4)	**<0.001**	1758 (59.8)	**<0.001**
Surgical wards	4 (7.1)	744 (15)	0.10	744 (25.3)	0.002
Pediatric wards	2 (3.6)	140 (2.8)	0.67	140 (4.8)	0.67
Days of admission, median (IQR)	10 (3.25, 23.5)	3 (2.0. 7.0)	**<0.001**	5 (2.0.10–0)	**<0.001**
Readmission, *n* (%) ^a^	47 (11.4)	298 (8.5)	0.54	234 (11.8)	0.95
Country of birth, *n* (%) ^b^					
Bolivia	9 (50)	705 (59.0)	0.69	431 (58.2)	0.49
Spain	3(16)	368 (30.8)	0.20	225 (30.4)	0.21
Ecuador	1 (5.6)	22 (1.8)	0.25	19 (2.6)	0.43
Paraguay	0 (0)	28 (0.6)	1.00	16 (2.2)	1
Argentina	0 (0)	21 (0.4)	0.57	18 (2.4)	1
Colombia	1 (5.6)	12 (1.0)	0.18	8 (1.1)	0.20
Venezuela	0 (0)	10 (0.8)	1.00	7 (0.9)	1
El Salvador	0 (0)	9 (0.8)	1.00	5 (0.5)	1
Brazil	1 (1.8)	6 (0.1)	1.00	6 (0.8)	0.16
Chile	0 (0)	4 (0.3)	1.00	3 (0.4)	1
Type of diagnostic, *n* (%)					
CD without organ involvement	20 (35.7)	2834 (57.1)	**0.001**	1368 (46.6)	0.11
CD with heart involvement	28 (50)	1483 (29.9)	**<0.001**	1211 (41.2)	0.19
CD with organ involvement	10 (17.9)	619 (12.5)	0.23	366 (12.5)	0.29
Trypanosomiasis non-specified	4 (7.1)	498 (10)	0.47	221 (7.5)	0.92

**^a^** available in 35 non-survivors and in 3450 survivors; **^b^** available in 18 non-survivors and in 1193 survivors HIV: human immunodeficiency virus; IQR: interquartile range. In bold, statistically significant differences.

**Table 4 microorganisms-09-01991-t004:** Bivariable and multivariable analysis of risk factors associated with mortality in hospitalized people with Chagas disease (CD).

Variables	Total Cases			Non-Obstetrics/Gynecology Cases		
	Crude OR (95% CI)	Adjusted OR (95% CI)	*p*-Value	Crude OR (95% CIs)	Adjusted OR (95% CI)	*p*-Value
Age	1.06 (1.04, 1.08)	1.05 (1.03, 1.07)	**<0.001**	1.04 (1.02, 1.06)	1.04 (1.02, 1.23)	**<0.001**
Sex, male	3.42 (2.20, 5.81)	1.79 (1.03, 3.14)	**0.039**	1.96 (0.95, 1.48)	1.40 (0.82, 2.41)	0.22
Neoplasms	6.97 (3.56, 13.65)	4.89 (2.13, 11.24)	**<0.001**	4.47 (2.27, 8.83)	4.46 (1.97, 10.1)	**<0.001**
Heart failure	2.79 (1.39, 5.57)	1.45 (0.67, 3.16)	0.34	1.57 (0.78, 3.71)	1.30 (0.81, 2.79)	0.69
Diabetes mellitus	3.67 (1.64, 8.21)	1.21 (0.48, 3.00)	0.68	2.16 (0.96, 4.84)	1.19 (0.49, 2.93)	0.69
Chronic kidney failure	5.98 (2.32, 15.40)	2.25 (0.80, 6.34)	0.12	3.59 (1.39, 9.25)	2.14 (0.77, 5.95)	0.15
HIV infection	12.07 (4.57, 31.84)	14.10 (4.88, 40.73)	**<0.001**	7.90 (2.97, 20.95)	10.46 (3.65, 30.0)	**<0.001**
Lymphoma/leukemia	5.34 (1.61, 17.68)	0.92 (0.21, 3.94)	0.92	3.14 (0.95, 10.3)	0.78 (0.18, 3.32)	0.73
CD with heart involvement	2.39 (1.39, 3.98)	1.57 (0.85, 3.91)	0.15	1.42 (0.84, 2.14)	0.94 (0.41, 2.15)	0.89
CD without organ involvement	0.41 (0.24, 0.72)	0.55 (0.26, 1.17)	0.13	0.63 (0.36, 1.10)	0.73 (0.31, 1.68)	0.46

CI: confidence intervals; OR: odds ratio. In bold, statistically significant differences.

## Data Availability

The final dataset is fully available online GitHub https://github.com/jramosrincon/CMBDChagas.git (accessed on 11 August 2021).

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
