# Peer review of "Chagas Disease-Related Mortality in Spain, 1997 to 2018"

_microorganisms, 2021, doi:10.3390/microorganisms9091991_

Round 1

Reviewer 1 Report

I assessed the manuscript entitled “Chagas disease-related mortality in Spain, 1997 to 2018”. The authors aimed to analyse mortality, causes of death and factors associated in patients admitted to Spanish hospitals covered by a national hospital discharge database. The authors performed a retrospective study using administrative deanonymized data of hospital discharge by selecting patients with Chagas disease classified using ICD-9 and ICD-10 according to the period of hospitalization. The dataset used for the analysis is far to be complete (as mentioned by the authors in the limitation section) to perform analysis to infer about mortality and factors associated to mortality in patients with Chagas disease. Overall, the topic is of some interest but the intrinsic limitations in the dataset limit in deep inference regarding Chagas disease-related death in hospitalized subjects.

Major comments:

- Title: the title does not represent what is investigated across the manuscript

-Methods: it seems that the impossibility of the authors to directly access to patient’s medical record to ascertain if the death was truly related to Chagas diseases exposed the study at high risk of bias (main objective and outcome at high risk of bias).

-Methods: it is unclear which is the clinical relevance of the multivariable model of factors associated with death in inpatients admitted with Chagas disease.

-Results: by looking at the trend of new Chagas disease admission there is a significant rise starting from 2006. This finding should be further discussed in light of improvement in policy of diagnosis and migratory routes. In particular the number of admission seems to be the more solid result when compared to all the other analysis and related inferred results of the manuscript.

Minor comments:

-Introduction (line 50): Chagas disease is not only a chronic disease such as stated thereafter into the introduction section.

-Discussion (line 238-239): the percentages reported are unclear

                    (line 263): (cita.)

Author Response

Authors reply to reviewer 1
I assessed the manuscript entitled “Chagas disease-related mortality in Spain, 1997 to 2018”. The authors aimed to analyse mortality, causes of death and factors associated in patients admitted to Spanish hospitals covered by a national hospital discharge database. The authors performed a retrospective study using administrative deanonymized data of hospital discharge by selecting patients with Chagas disease classified using ICD-9 and ICD-10 according to the period of hospitalization. The dataset used for the analysis is far to be complete (as mentioned by the authors in the limitation section) to perform analysis to infer about mortality and factors associated to mortality in patients with Chagas disease. Overall, the topic is of some interest but the intrinsic limitations in the dataset limit in deep inference regarding Chagas disease-related death in hospitalized subjects.
Major comments:
- Title: the title does not represent what is investigated across the manuscript
Authors' reply:

Following the reviewer's suggestions, we have modified the title, from ‘Chagas disease-related mortality in Spain, 1997 to 2018’ to ‘Chagas disease-related in-hospital mortality in Spain. A cross-sectional analysis of the Spanish National Hospital Discharge Database from 1997 to 2018’.
-Methods: it seems that the impossibility of the authors to directly access to patient’s medical record to ascertain if the death was truly related to Chagas diseases exposed the study at high risk of bias (main objective and outcome at high risk of bias).
Authors' reply:
We agree that this is a limitation, as the Spanish National Hospital Discharge Database does not allow us to review the 56 deaths, and we only have the information from the database. We have considered criteria adapted to the database to define whether the mortality was possibly related to CD. In this, our methods are similar to other studies in endemic countries that have also used the national registry based on ICD-9 and ICD-10. 
-Methods: it is unclear which is the clinical relevance of the multivariable model of factors associated with death in inpatients admitted with Chagas disease.
Authors' reply,  

Following the reviewer's suggestion, we have added the variables included in the multivariable model and have eliminated the concept of clinical relevance to avoid confusion. 
Previously: “Variables yielding a p value of less than 0.1 in the univariable analysis, plus clinical and biological variables with a plausible association to the outcome, were entered into a multivariable logistic regression, using a stepwise selection method with the likelihood ratio test.”
Now: “Demographic variables, comorbidities, and diagnostic variables yielding a p value of less than 0.1 in the univariable analysis were entered into a multivariable logistic regression, using a stepwise selection method with the likelihood ratio test.”
-Results: by looking at the trend of new Chagas disease admission there is a significant rise starting from 2006. This finding should be further discussed in light of improvement in policy of diagnosis and migratory routes. In particular the number of admission seems to be the more solid result when compared to all the other analysis and related inferred results of the manuscript.
Authors' reply,  

We agree with the reviewer that there is a trend of new admissions for Chagas disease, a significant increase is observed from 2006 onwards. We have not discussed it because it is a topic that has been addressed in a previous article using the same database, and we focused on mortality with Chagas disease. While we have discussed analysis and inferred results related to the manuscript, we did not mention this point. Now we have  added a new paragraph and 6 references.

New: “Moreover, CD admissions show a significant upward trend in Spain starting from 2006. This is due to a sharp increase in immigration from Latin America to Spain, which started in 1997 but accelerated from 2005 to 2010, especially in women of childbearing age. Migration in women of childbearing age from Latin America may spread CD to non-endemic areas through vertical transmission [40]. If CD is diagnosed during pregnancy, the transmission of T. cruzi to the child cannot be prevented, so early diagnosis in newborns is essential for administering appropriate treatment [40,41]. In Spain, serological screening of T. cruzi infection in pregnant women from Latin American is not systematically carried out, but depends on each autonomous community; some regions have implemented prenatal screening programs since 2007 [42,43]. Moreover, a community-based intervention implemented in Spain has shown to be effective in providing access to Chagas disease diagnosis, which has increased in recent years [44,45].”
40.Ramos JM, Milla A, Rodríguez JC, López-Chejade P, Flóres M, Rodríguez JM, et al. Chagas disease in Latin American pregnant immigrants: experience in a non-endemic country. Arch Gynecol Obstet. 2012;285: 919–23. doi:10.1007/s00404-011-2081-9
41. Llenas-García J, Wikman-Jorgensen P, Gil-Anguita C, Ramos-Sesma V, Torrús-Tendero D, Martínez-Goñi R, et al. Chagas disease screening in pregnant Latin American women: adherence to a systematic screening protocol in a non-endemic country. PLoS Negl Trop Dis.2021;15:e0009281. doi: 10.1371/journal.pntd.0009281
42.Generalitat. Conselleria de Sanitat, editor. Enfermedad de Chagas Importada. Protocolo de Actuación en la Comuninat Valenciana. Valencia; 2009. 
43.Programa de prevención y control de la enfermedad de Chagas congénita en, Cataluña, Subdirección General de Vigilancia y Respuesta a Emergencias de Salud Pública. Protocolo de cribado, diagnótico  tratamiento de la Enfermedad de Chagas en mujeresembarazadas latinoamericanas y en sus hijos. Agencia de Salud Pública de Cataluña. Departamento de Salud, editor. Barcelona; 2018. 
44.Romay-Barja M, Iglesias-Rus L, Boquete T, Benito A, Blasco-Hernández T. Key Chagas disease missing knowledge among at-risk population in Spain affecting diagnosis and treatment. Infect Dis Poverty. 2021;10:55. doi: 10.1186/s40249-021-00841-4.1).
45. Ramos-Sesma V, Navarro M, Llenas-Garcia J, Gil-Anguita C, Torrus-Tendero D et al. Community-based screening of Chagas disease among Latin American migrants in a non-endemic country: an observational study. Infect Dis Poverty. 2021
.

Minor comments:
-Introduction (line 50): Chagas disease is not only a chronic disease such as stated thereafter into the introduction section.
Authors' reply,  
We have modified the text to “… acute and chronic systemic parasitic infection”
-Discussion (line 238-239): the percentages reported are unclear
Authors' reply, Following the reviewer's suggestions, we have rewritten the sentence
Previously: “Of the deaths recorded in Colombia from 1979 to 2018, 0.04% were related to CD, and 86.3% had an underlying cause.”
New: “Of the 7,287,461 deaths recorded, 3,276 (0.04%) were related to CD: 2,827 (86.3%) as an underlying cause and 449 (13.7%) as an associated cause.”
                    (line 263): (cita.)
Authors' reply:  
We have supplied the correct reference.

Reviewer 2 Report

After a detailed evaluation, analyzing the scientific relevance of these interesting findings, this manuscript should be accepted in the current form. The text is very clear and well-written. The authors pointed to the importance of this first study in non-endemic country. For sure, it is very relevant, and could be the basis for novel more detailed studies in the future. 

Author Response

Authors reply to Reviewer 2.

After a detailed evaluation, analyzing the scientific relevance of these interesting findings, this manuscript should be accepted in the current form. The text is very clear and well-written. The authors pointed to the importance of this first study in non-endemic country. For sure, it is very relevant, and could be the basis for novel more detailed studies in the future.

Authors' reply, Thanks for the commentary

Round 2

Reviewer 1 Report

Thank you for the revision  of the manuscript.